# Modelling the drivers of outbreak communication in online media news for improved event-based surveillance

Solene Rodde[1,2], Pachka Hammami[1,2], Asma Mesdour[1,2], Sarah Valentin[1,3], Bahdja Boudoua[3,4], Paolo Tizzani[5], Lina Awada[5], Carlene Trevennec[2,4], Paulo Pimenta[6], Andrea Apolloni[1,2], Elena Arsevska[1,2]*

1 French Agricultural Research Centre for International Development (CIRAD), Montpellier, France, 2 Joint Research Unit Animals, Health, Territories, Risks, and Ecosystems (UMR ASTRE), French Agricultural Research Centre for International Development (CIRAD), National Research Institute for Agriculture, Food and Environment (INRAE), Montpellier, France, 3 Joint Research Unit Land, Environment, Remote Sensing and Spatial Information (UMR TETIS), French Agricultural Research Centre for International Development (CIRAD), National Research Institute for Agriculture, Food and Environment (INRAE), Montpellier, France, 4 National Research Institute for Agriculture, Food and Environment (INRAE), Montpellier, France, 5 Data Integration Department, World Organisation for Animal Health (WOAH), Paris, France, 6 Laboratoire d'Informatique, de Robotique et de Microélectronique de Montpellier, Microelectronics Department, Montpellier, France

☙ These authors contributed equally to this work.
* elena.arsevska@cirad.fr

**Data availability statement:** The code and data needed to reproduce the results of this work are

## Abstract

Epidemic intelligence (EI) practitioners at health agencies monitor various sources to detect and follow up on disease outbreak news, including online media monitoring. The Platform for Automated Extraction of Disease Information from the Web (PADI-web), developed in 2016 for the French Platform for Epidemiosurveillance in Animal Health (Platform ESA), monitors and collects outbreak-related news from online media, allowing users to detect and anticipate response to disease outbreaks. Given the mass number of outbreak-related news collected with PADI-web, we aimed to understand better what drives communication on outbreaks by the different online media sources captured by this tool to allow for a more targeted and efficient EI process by its users. We built a bipartite network of sources communicating on outbreaks of avian influenza (AI) and African swine fever (ASF) captured by PADI-web between 2018 and 2019 worldwide. We used an Exponential Random Graph Model (ERGM) to assess epidemiological, socioeconomic, and cultural factors that drive communication on disease outbreaks from the different online media sources. Our AI network comprised 969 communicated news (links) from 436 news reports from 212 sources describing 199 AI outbreaks. The ASF network comprised 1340 communicated news (links) from 594 news reports from 204 sources and 277 ASF outbreaks. The ERGM was fitted for each network. In both models, international organisations and press agency sites were more likely to communicate about outbreaks than online news sites (OR = 4.8 and OR = 3.2, p < 0.001 for AI; OR = 3.1 and OR = 4.7, p < 0.001 for ASF). Research organisations for AI (OR = 2.3, p < 0.001) and veterinary authorities for ASF (OR = 3.6, p < 0.001) were also more likely

available at https://github.com/arsevska/ergm_padi_web and CIRAD Dataverse database at https://doi.org/10.18167/DVN1/W858OB.

**Funding:** H2020 MOOD project Grant agreement ID: 874850: https://cordis.europa.eu/project/id/874850. This study was also funded by the Direction générale de l'alimentation (DGAL), France. The funders had no role in study design, data collection and analysis, decision to publish, or preparation of the manuscript.

**Competing interests:** The authors have declared that no competing interests exist.

to be a source of information than online news sites. Our work identified the factors driving communication about animal and zoonotic infectious disease outbreaks in online media sources monitored by PADI-web. This information can guide EI practitioners and users of PADI-web to monitor specific sources based on their specialisation and coverage and the disease's epidemiological status. Our results also suggest that EI practitioners may use other means to collect EI information in countries and regions that are not well-represented in the data.

## Introduction

Epidemic intelligence (EI) is an essential tool for detection, verification and alerting for a rapid response towards rumours and confirmed outbreaks of emerging infectious diseases [1,2]. Traditionally, EI relies on the assessment of data collected from traditional surveillance, i.e., clinical visits by an animal/health practitioner, laboratory confirmation of the suspected pathogen, data centralisation at local and national level at public/animal health (PH/AH) authorities up to notification to international organisations, such as the World Organisation for Animal Health (WOAH), in the case of mandatory notifiable diseases in animals [1,3]. Consequently, EI based on data from traditional, indicator-based surveillance (IBS) could impact the accuracy of risk assessment due to the delayed or under-reported number of cases or outbreaks, depending, among others, on the country-level preparedness [4,5].

Since the early 2000s, online media news has been a new way to communicate about health events, including disease outbreaks. According to some estimates, about 65% of the world's first news about infectious disease events comes from informal sources, including press reports and the internet [2]. Thus, event-based surveillance (EBS) tools for online media news monitoring have shown promise in complementing traditional IBS sources for a more efficient EI process [6–9].

The Platform for Automated Extraction of Disease Information from the Web (PADI-web) was developed in 2016 for the International Animal Health Monitoring Unit of the French Platform for Epidemic Surveillance in Animal Health (ESA) [10–12]. PADI-web collects real-time data from Google News web aggregator in several languages, uses data mining algorithms to keep relevant disease-outbreak news, and extracts information on the disease, host, date and location of events mentioned in the news in a structured format accompanied by a visual interface (https://padi-web.cirad.fr/en/) and email alerts set by users [11]. Between 2016 and 2019, PADI-web collected over 70,000 news articles for several animal diseases, such as avian influenza (AI) and African swine fever (ASF), which are diseases of importance to the French Platform ESA [11].

In previous work, we explored the information flow, i.e., who cites who in the online media news about AI outbreaks occurring globally between 2018 and 2019, obtained through PADI-web [13]. We characterised the role of the different sources in the early vs late communication, i.e., before vs after official notification of an outbreak to the World Organisation for Animal Health (WOAH). The early news on AI outbreaks mainly came from non-official sources such as press agencies but also national authorities. In contrast, the late news relied primarily on official sources, such as WOAH [13].

We built upon our previous work from Valentin et al., 2023 [13] and exploited the existing dataset from PADI-web, i.e., the network of sources from the news on AI outbreaks and extended to an additional disease, ASF. We used network analysis to:

i) assess what drives the communication on outbreaks among online media sources given the different epidemiological, socio-economical and cultural factors associated with the country and region of the outbreaks and the characteristics of the sources,

ii) compare if there are differences in outbreak communication between diseases.

We used AI and ASF as disease models, as they are highly infectious pathogens with sanitary and production impacts and represent contrasted examples of purely animal (ASF) and zoonotic (AI) diseases. Early detection of their emergence is crucial for assessing the risk of introduction in France. Outbreaks of these two diseases are actively monitored globally by the French International Animal Health Monitoring Unit of the Platform ESA.

This paper is organised as follows: We first present the creation of the bipartite network of sources and outbreaks, including the associated covariates. We further describe the exploratory-descriptive and analytical-modelling approach to the bipartite network. Finally, we present and discuss the study results, limitations, and future perspectives.

## Materials and methods

### Data and construction of the bipartite network

Data on the two diseases, AI and ASF, were collected from PADI-web news articles from July 1, 2018, to June 30, 2019. This one-year timeframe was chosen to capture the global spatiotemporal patterns of the two diseases' epidemics and provide a clear basis for modelling and completing our previous work [13]. The dataset included two components: outbreaks and sources.

Outbreak data consisted of events reported in the news, detailing the date, location, and host of each occurrence. These events were matched with official outbreaks reported by countries to the WOAH. We manually matched the events based on several criteria: disease, location, date of event and host (199 matched for AI, 277 for ASF). Events mentioned in the news that could not be matched to official data (31 for AI, 69 for ASF) were excluded as we could not verify the truthfulness of the described outbreaks in the news.

Sources were manually identified by labelling all entities in the news that served as information on a given outbreak (e.g., Reuters press agency, CIDRAP research centre, etc.). Links were established between sources and the outbreaks they described, revealing 212 unique sources for AI and 204 for ASF.

We created a bipartite (two-mode) network with two nodes of different natures: nodes for sources and nodes for outbreaks (Fig 1) for each disease, respectively. We decided to use a bipartite network because of the nature of our data: two distinct sets of objects of different natures (sources and outbreaks) linked by information flows, forming a complex and asymmetric network. In this type of network, links only connect nodes of different types: sources and outbreaks [14]. Each outbreak is connected to at least one source. The link (edge) between the two modes corresponded to a communicated outbreak news from a given source, and multiple edges starting from an outbreak indicate that different sources had reported news on the respective outbreak. Covariates were implemented in the network as node attributes.

Source covariates were categorized per i) type (international organisation, company/association, laboratory, official authority, veterinary authority, online news, press agency, radio/TV, social platform, research organisation; ii) geographical focus (international, national, local) and iii) specialisation (general vs specialised health-oriented) (Fig 1). A detailed description of these source attributes is provided in our previous work [13].

Outbreak covariates were related to the epidemiological, socio-economical and cultural context of the country of the outbreaks (Fig 1).

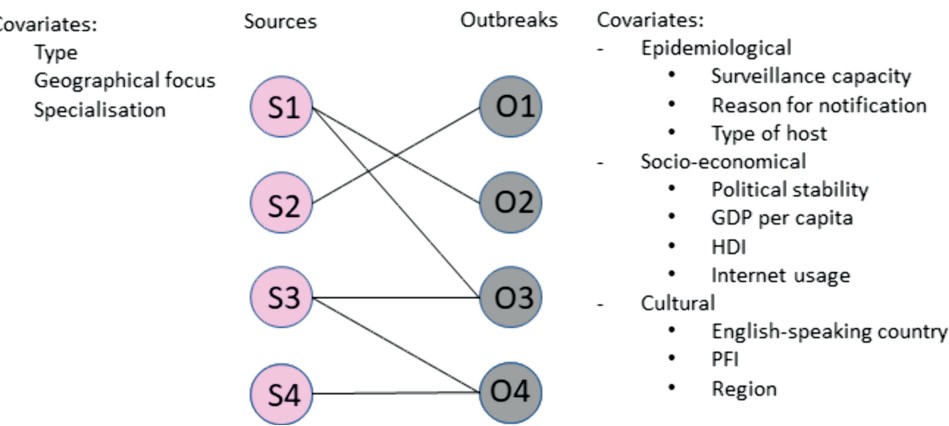

**Fig 1. Schema of the bipartite network of sources (S) and outbreaks (O) and their associated covariates. Gross Domestic Product (GDP); Human Development Index (HDI); Press Freedom Index (PFI)**

The epidemiological covariates used were: i) Surveillance capacity of the country, an indicator based on a country's self-assessment informing its capacity to detect, assess, notify, report and respond to public health risks and acute outbreaks of national and international concern (ranging from 0 to 100) [15]; ii) Reason for notification to WOAH: first occurrence in a zone/country, recurrence of the disease, new strain and unknown reason, in case there was no information available [16]; iii) Type of host, where the domestic pig and poultry were considered domestic; the remaining were wild or environmental if the sample came from the environment [16].

The socio-economical covariates per country and year were: i) Political stability, an estimate measuring perceptions of the likelihood of political instability and politically motivated violence, including terrorism, (ranging from -2.5 to 2.5; lower the value, lower the stability) [17]; ii) Gross Domestic Product (GDP) per capita [17]; iii) Human Development Index (HDI) [18]; iv) Internet usage expressed by the percentage of the population using the internet [17].

The following covariates expressed the cultural context at the country level: i) English-speaking country ('yes' if English is an official language, or 'no'); ii) Press Freedom Indicator (PFI), an indicator developed by RSF (Reporter Sans Frontière) which measures the degree of press freedom enjoyed by journalists and media in a country on a scale of 0 to 100 [19]; iii) Region (African Region [AFR], Region of the Americas [AMR], Southeast Asian Region [SEAR], European Region (EUR), Eastern Mediterranean Region [EMR], Western Pacific Region [WPR]) [20].

All covariates associated with the outbreaks were collected and standardised per country and year, except for the reason of notification and type of host, which corresponded to each outbreak.

## Descriptive analysis

An exploratory and descriptive analysis was implemented at a global and node level for the AI and ASF networks.

**Global level**. We assessed the network density, i.e., the ratio of the number of edges to the total number of possible edges. We also checked for the network components (weakly vs.

strongly connected components). Components are portions of the network that are disconnected from each other. An isolated component is a subset of nodes that are not connected to other nodes in the network.

**Node level**. We calculated the degree of each node in the network (for sources and outbreaks, respectively), i.e., the number of connections it has to other nodes. The degree distribution was the probability distribution of these degrees over the whole network. More precisely, the source degree showed the number of times a given source communicated about an outbreak, while the degree of an outbreak was the number of times a given outbreak was communicated in the news.

### Exponential random graph model

Before modelling, we used the Pearson correlation test to check for a potential correlation between continuous quantitative covariates (Fig 2). Only variables with a correlation coefficient between -0.7 and 0.7 were retained for further analysis.

Then, we used a bipartite exponential random graph model (ERGM) to identify the factors that influence the establishment of links between sources and outbreaks and, more precisely, to assess what drives a source to communicate on a given outbreak and later compare between diseases.

ERGM models (either applied to mono or bipartite networks) are well suited to study how nodes and links' characteristics contribute to the formation of a network. The approach takes into account the interdependencies among nodes. It constrains the network to have certain global and centrality features, something that simple regression does not account for the structure or dependencies between multiple types of nodes [21–23]. Using a Bayesian approach, the space of all possible configurations is explored, and the probability of link formation is inferred from the frequencies of network occurrences. The coefficient analysis

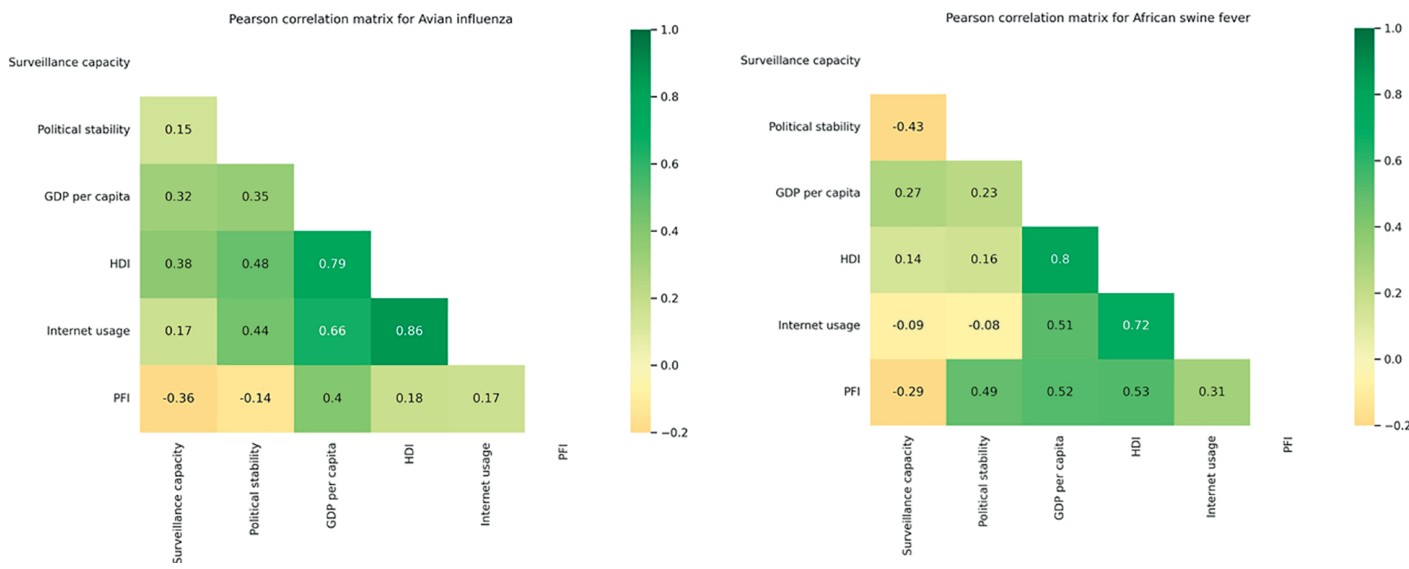

**Fig 2. Pearson correlation plot for the quantitative covariates.** Gross Domestic Product (GDP); Human Development Index (HDI); Press Freedom Index (PFI).

would provide a better understanding of the effects of node characteristics on the creation of links (interpreted here as communication from a source for an outbreak) [24].

Model fits were only considered when ERGM fitting stopped after the first estimate (using maximum pseudo-likelihood estimates, MPLE) or when maximum likelihood estimation (MLE) converged without generating a warning (using a Markov Chain Monte Carlo (MCMC) estimation algorithm with a Metropolis-Hastings sampler. The ERGM model structure and parameters were selected by sequential tests, following a simple stepwise procedure to identify the most parsimonious parameter combination, as described in Hammami et al., 2022 [23]. Starting with a prospective approach from the simplest model, including only links as a predictor, the addition of each network statistic was tested sequentially to identify a relevant subset of predictors (network statistics) based on the Akaike Information Criterion (AIC) indicator. Then, a retrospective approach was used, sequentially removing previously selected predictors from the model to retain only those improving model fit.

**Model validation**. In-sample performance was assessed for the ERGM selected for each disease by a goodness-of-fit (Gof) analysis comparing three structural statistics of the observed network to those of networks randomly simulated by the adjusted ERGM (minimum geodesic distance, distribution of degree per type of node) [25]. One thousand networks were randomly simulated to achieve the Gof for the network models. The statistics compared were the degree of source nodes, degree of outbreak nodes and minimum geodesic distance.

**Software**. The analysis was carried out using the R programming language (version 4.3.1) [26], and the associated packages: "igraph" (version 2.0.1.1) [27], "network" (version 1.18.2) [28], "ergm" (version 4.6.0) [29] and "bipartite" (version 2.19) [30]. Data and code supporting the results of the work are available in Data and code availability section.

**Ethics approval**. Ethics approval was not required for this study as it includes information (news) freely available in the public domain. This research does not contain any studies with animals or humans performed by any of the authors.

## Results

### Descriptive analysis

**Avian influenza.** The AI network comprised 212 sources, 199 outbreaks, and 969 links. The network had low density (0.023), indicating low node connectivity.

Based on the number of connections, the most important sources were the WOAH (deg = 140), Center for Infectious Disease Research and Policy (CIDRAP News) (deg = 129) and WATTAgNet online news (deg = 59).

The AI outbreaks were from six regions: Europe, Western Pacific, Southeast Asia, Eastern Mediterranean, America and Africa (Figs 3 and 4). The majority of AI outbreaks were in Chinese Taipei (n = 38; 19.1%), Bulgaria (n = 16; 8.04%) and Mexico (n = 15; 7.54%). The highest degree was related to three outbreaks in India (deg = 36, deg = 22, deg = 20, respectively), followed by outbreaks in the United States of America (deg = 20) and Malaysia (deg = 19) (Supp file 2).

**African swine fever.** The ASF network comprised 204 sources, 277 outbreaks and 1340 links (Supp file 1). The network had low density (0.024), indicating low node connectivity.

Based on the number of connections, the most important sources were the Reuters Press Agency (deg = 158), China Veterinary Authority (deg = 125), and the Xinhua Press Agency (deg = 87).

The ASF outbreaks were from four regions: Europe, Western Pacific, South-East Asia and Africa (Figs 3 and 4). The majority of ASF outbreaks were in China (n = 120; 43.32%), followed by Vietnam (n = 41; 14.8%) and Bulgaria (n = 30; 10.83%). Among the outbreaks with

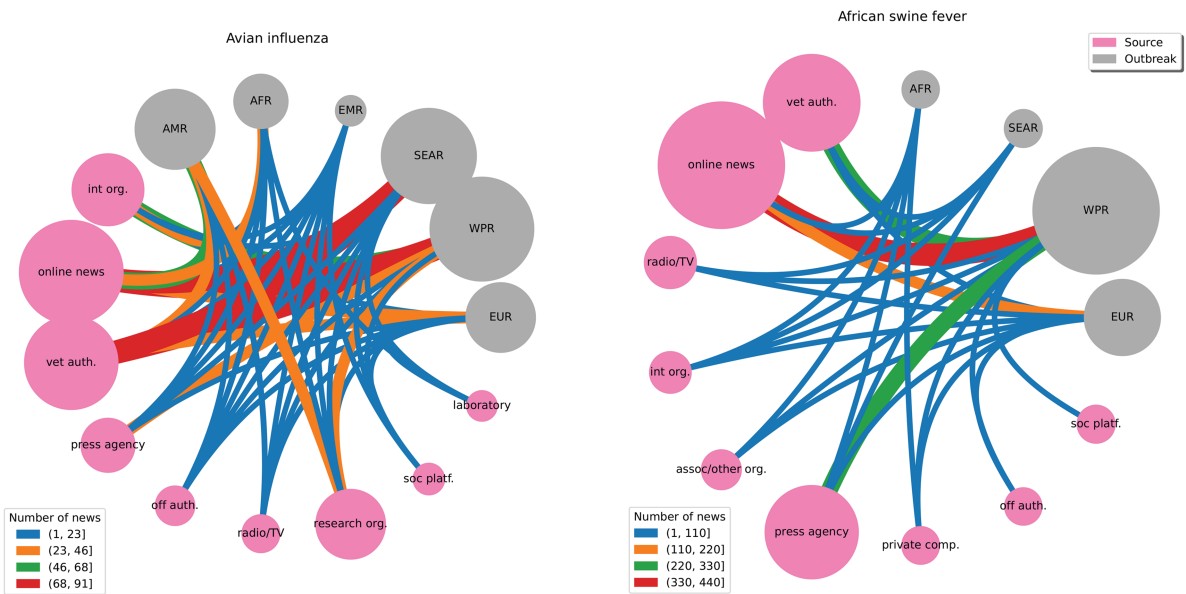

**Fig 3. Chord diagram of links between the sources and outbreaks of avian influenza (left) and African swine fever networks (right).** The node size corresponds to the number of sources per type and outbreaks per region. The edge width corresponds to the number of links (communicated news on outbreaks) between a source type and region: AMR, Region of the Americas; AFR, African Region; EMR: Eastern Mediterranean Region; SEAR, Southeast Asian Region; WPR, Western Pacific Region; EUR: European Region.

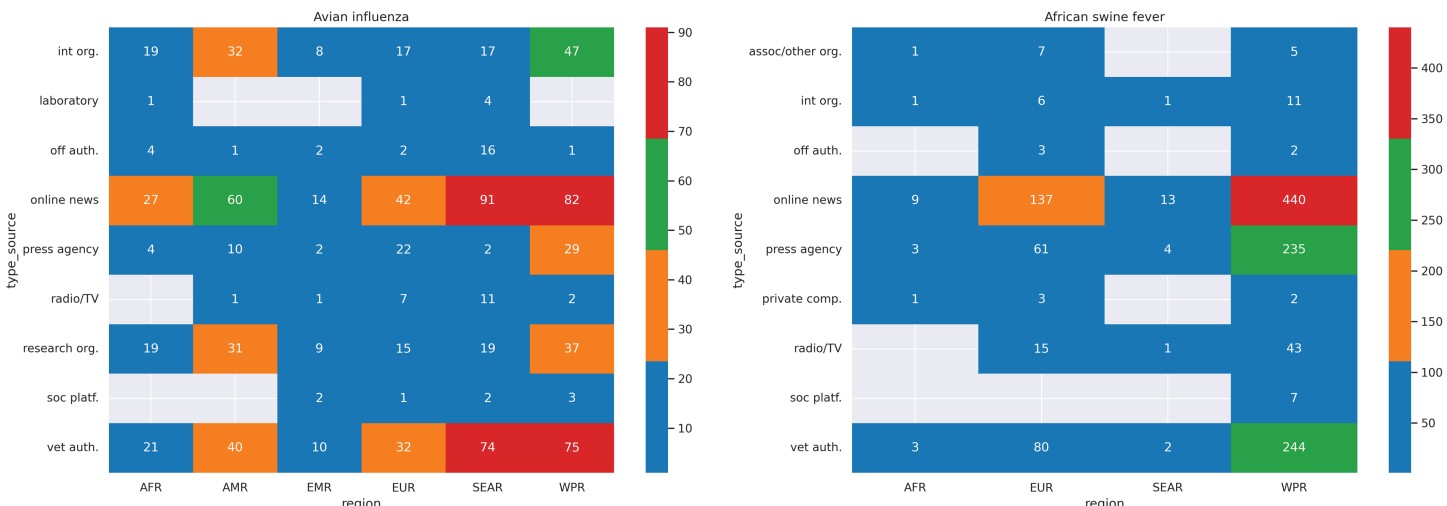

**Fig 4. Heatmap of the number of communicated news on outbreaks per source type and region.** Regions: AMR, Region of the Americas; AFR, African Region; EMR: Eastern Mediterranean Region; SEAR, Southeast Asian Region; WPR, Western Pacific Region; EUR: European Region.

the highest degree, one occurred in China (deg = 24), one in the Democratic People's Republic of Korea (North Korea) (deg = 21), one in Slovakia (deg = 16) and two in Belgium (deg = 15, deg = 14, respectively) (Supp file 2).

## Exponential random graph model

Table 1 shows the key factors for information dissemination on AI and ASF outbreaks in online media selected using the stepwise procedure.

**Avian influenza**. The AI model obtained with a stepwise selection procedure comprised of eleven covariates (Table 1, AIC of 7175). Based on the odds ratio (OR), the most important covariates involved in the likelihood of a link between sources and AI outbreaks were the source type, the geographical focus, the specialisation of the source, and the region and language of the country where the AI outbreaks occurred (Fig 5).

> **Type of source.** International organisations, research organisations and press agencies had the highest probability of reporting on AI outbreaks compared to online news sources (OR = 4.82, OR = 2.29, OR = 3.17, p < 0.001, respectively; Fig 5). Social platforms and veterinary authority sources were less likely to communicate about AI outbreaks than online news (p < 0.001; Fig 5).
>
> **Geographical focus and specialisation of the source.** International and national coverage sources were more likely to communicate about outbreaks of AI than local sources (OR = 2.99, OR = 1.95, p < 0.001, respectively; Fig 5). Specialised sources were 4.4 times more likely to communicate information on AI outbreaks than generalised sources (p < 0.001; Fig 5).
>
> **Region of the outbreaks.** AI outbreaks in Europe, America and Southeast Asia were more likely to be communicated by the sources (OR = 2.57, OR = 9.95, OR = 2.89, p < 0.001, p = 0.002, p < 0.001, respectively; Fig 5).
>
> **Host type.** AI outbreaks in wild birds were more likely to be communicated by the sources than outbreaks affecting domestic birds (OR = 1.67, p < 0.001; Fig 5).
>
> **Official language.** AI outbreaks occurring in a country where English is the official language were more likely to be communicated by the sources than outbreaks in non-English-speaking countries (OR = 1.76, p < 0.001; Fig 5).

**African swine fever**. The ASF model obtained with a stepwise selection process comprised six covariates (Table 1, AIC of 11789), with the most important covariates being the type and geographical focus of the source and the region of the outbreaks (Fig 6).

> **Type of source.** Similarly to the AI model, sources such as international organisations and press agencies were more likely to form links with outbreaks compared to online news sources (OR = 3.08, OR = 4.71, respectively, p < 0.001; Fig 6). However, compared

**Table 1. Covariates retained (shown as X) with a stepwise selection procedure for Avian influenza (AI) and African swine fever (ASF). Gross Domestic Product (GDP), Press Freedom Index (PFI).**

| Type node | ERGM terms | Covariate | AI | ASF |
|---|---|---|---|---|
| Source | nodefactor | Type source | X | X |
| | | Geographical focus | X | X |
| | | Specialisation | X | |
| Outbreak | nodefactor | Type host | X | |
| | | Political stability | X | |
| | | GDP per capita | X | X |
| | | Internet usage | X | X |
| | | Official language | X | |
| | | Region | X | X |
| | | PFI | X | X |
| | nodematch | Region (homophily) | X | |

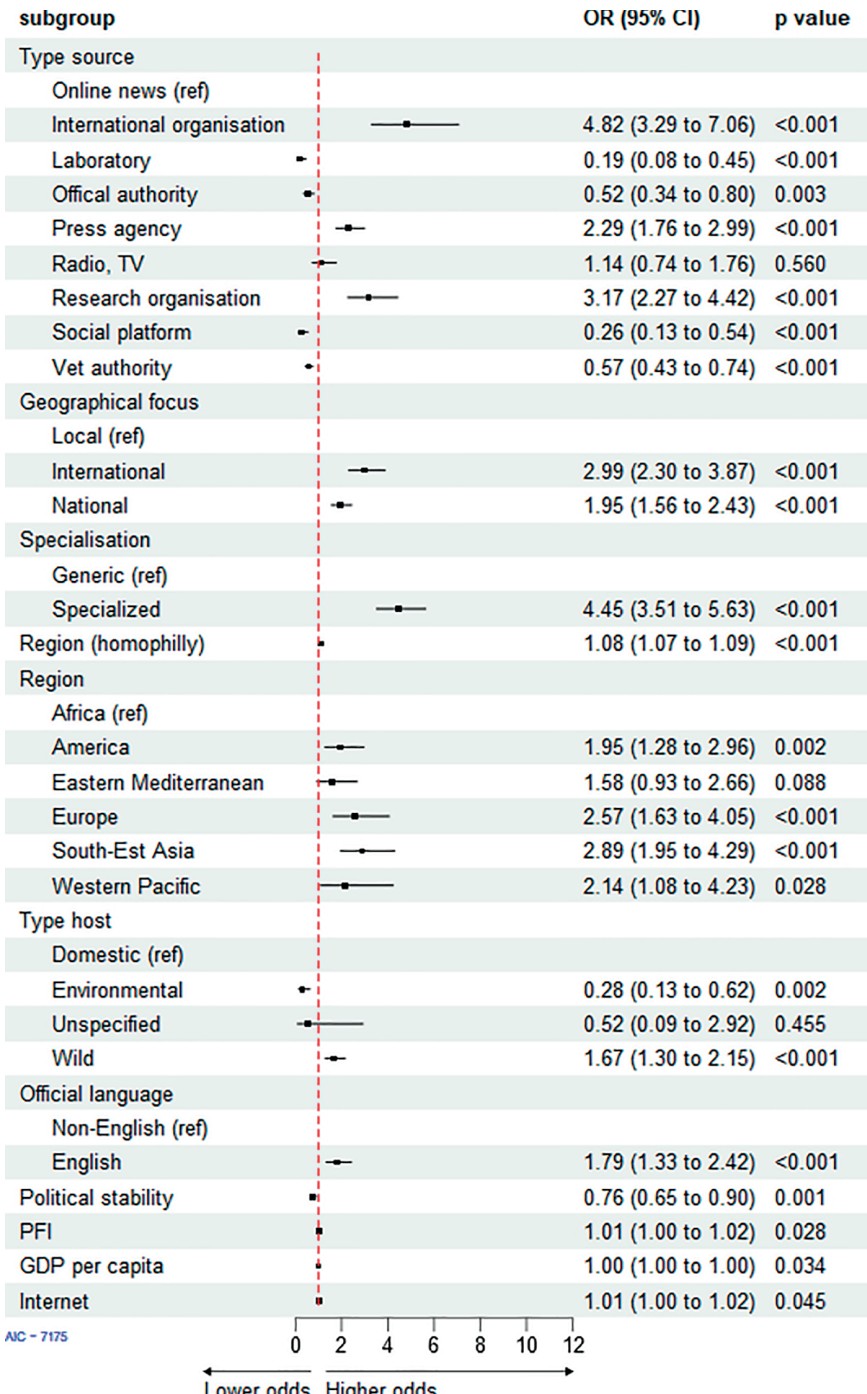

**Fig 5. Exponential random graph model (ERGM) results the bipartite network of sources and Avian influenza outbreaks.** OR = Odds ratio, CI = Confidence interval. Press Freedom Index (PFI), Gross Domestic Product (GDP).

to the AI model, the veterinary authorities were more likely to communicate about ASF outbreaks compared to online news sources (OR = 3.64, p < 0.001; Fig 6).

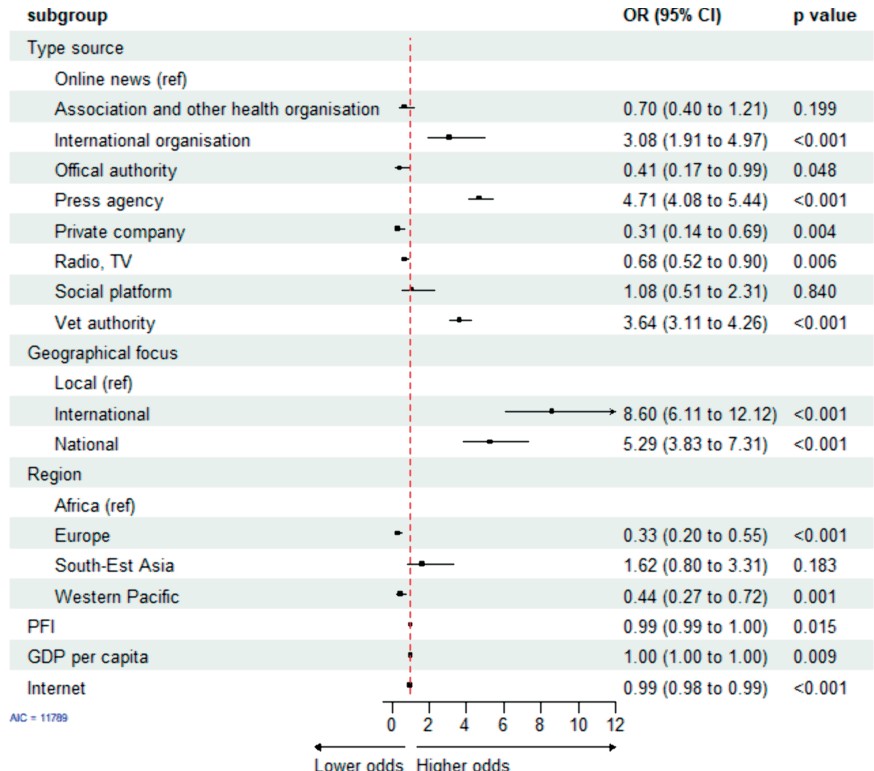

**Fig 6. Exponential random graph model (ERGM) results of the bipartite network of sources and African swine fever outbreaks.** OR = Odds ratio, CI = Confidence interval. Press Freedom Index (PFI), Gross Domestic Product (GDP).

**Geographical focus of the source.** Similarly to the AI model, international and national coverage sources were more likely to communicate about outbreaks of ASF than local sources (OR = 8.60, OR = 5.29, p < 0.001, respectively; Fig 6).

The Gof statistics showed a good model fit for both the AI and ASF models (Supp file 3), with a slightly better fit for the AI model.

## Discussion

This study assessed the factors influencing the communication of AI and ASF outbreaks in online media sources monitored by PADI-web. Our results show that outbreak communication in online media differs between diseases, depending on the region where the outbreaks occur, the hosts, and the source of information, their specialisation in the animal health domain and geographical coverage, including the official language(s) in the country of the outbreaks.

### News source types and specialisation for outbreak communication

Our findings suggest the important role of specialised sources in animal health, such as the national authorities, international and research organisations and press agencies for outbreak communication in online media, also shown in our related work [13]. In the case of AI, news coverage in our dataset predominantly referenced WOAH and CIDRAP, reflecting

its zoonotic nature and relevance to both animal and public health. Conversely, ASF is a non-zoonotic disease with significant economic implications for pig production; accordingly, online media reporting primarily relied on WOAH, national veterinary authorities, and press agencies, with minimal engagement from public health-focused sources.

Specialised and official sources are adept at conveying targeted information to professionals who should be informed about disease risks, preventive measures and control strategies that have been taken during disease outbreaks [31]. For example, the consistent identification of WOAH as a source of information for both ASF and AI outbreaks suggests a positive outcome regarding outbreak information dissemination and transparency. This may indicate progress toward WOAH's stated objective of fostering a robust network of stakeholders and accessible resources to facilitate effective information sharing, as outlined on its official website (https://www.woah.org/en/what-we-do/). In addition, the press and media play (such as Reuters and Xinhua News in our dataset) a key role in raising public awareness and shaping perceptions and understanding of risks to a broader audience besides professionals [32–35]. This approach ensures that EI practitioners and the general public, including concerned stakeholders (like farmers and industry), remain well-informed, which helps manage sanitary responses and curtail the spread of misinformation and panic [36–38].

In this context, the added value of media monitoring is to provide EBS capabilities by aggregating data from diverse sources, including reliable, official, and specialised outlets. This enables early intervention and more nuanced risk assessment from EBS sources [31], even when health authorities are aware of the event. For example, in the early months of the COVID-19 pandemic, EI from EBS at the World Health Organisation (WHO) complemented official COVID-19 case and death reporting, prompting verification processes for a more timely response. As the pandemic progressed, EI activities at the WHO provided a more comprehensive understanding of the situation, which could not be captured by a single type of surveillance alone [31].

## Timing and dynamics of outbreak communication in the online news

Previous studies have demonstrated that the online news environment is highly dynamic and context-dependent, particularly during an epidemic or when an outbreak affects new territories, countries, or hosts [36,39]. This variability in news reporting across epidemic phases is crucial for EI practitioners and risk assessors, as it directly influences their ability to monitor, assess, and respond to emerging threats. During the early stages of an outbreak, news reporting tends to focus on alerting the public and health authorities to a new threat, often with high uncertainty and rapid communication; in contrast, once the epidemic is established, media coverage shifts towards providing ongoing updates, analysis, and risk communication, which is critical for informing control measures, resource allocation, and public health decisions [36,40,41].

For example, online media monitored by PADI-web in 2018 frequently reported on the first introduction of ASF in China [42], as follows: Reuters - August 3, 2018 - China issued an African swine fever outbreak alert after the nation's first case was reported near the northeastern city of Shenyang, the Ministry of Agriculture and Rural Affairs said on Friday (article id 9019aa6d0e from "asf reports" dataset, see Code and Data availability statement). PADI-web news also informed on the consequent ASF epidemics in the country, as follows: Feedstuffs - August 30, 2018 - A new outbreak of African swine fever (ASF) may have been discovered in the city of Wuhu, in eastern China's Anhui province, China's Ministry of Agriculture and Rural Affairs reported through media outlets in the country (article id SEO944JPSL from "asf reports" dataset, see Code and Data availability statement).

Although we did not examine the drivers of outbreak communication at different epidemic stages or before/after the official notification to WOAH [41,43], we plan to address these factors in future research. Furthermore, in the current dataset, 52 AI outbreaks (26.13% of total AI outbreaks) and 99 ASF outbreaks (35.74% of total ASF outbreaks) were reported in online news before WOAH's official notification. Analysis of these corpora revealed that outbreaks that occur within the same region (region homophily), wild host outbreaks, and higher HDI of the country of the outbreaks were significantly associated ($p < 0.05$) with faster communication of AI outbreaks. For ASF, higher HDI of the country of the outbreaks, press agencies, national veterinary authorities, and sources with more extensive geographical coverage were significantly associated ($p < 0.05$) with early outbreak communication. While these data subsets were insufficient to draw definitive conclusions, expanding the dataset to include more recent years and larger-scale epidemics could reveal new patterns or confirm our preliminary findings regarding factors for outbreak communication in online media.

## Region of the outbreaks and news coverage of online media

Our analysis revealed that AI outbreaks across all regions were more frequently reported in online news than outbreaks in the African region. Additionally, while lower, the influence of region homophily on AI outbreak communication was also significant.

Region homophily in the context of outbreak communication refers to the tendency for news and sources to report more frequently on outbreaks within geographically same regions. News sources in a given region are more likely to report on outbreaks that occur locally or in nearby areas due to the perceived relevance to their audience. This could be because the outbreak directly impacts local communities, farmers or the economy, raising awareness and urgency [44]. Additionally, regions close to an outbreak zone may prioritise coverage due to concerns about cross-border transmission, prompting local authorities and media to alert populations and mitigate risks. Finally, regional or national media often have established networks, sources, and contacts in their geographic areas. This results in faster and more frequent coverage when local or neighbouring regions experience outbreaks.

On the other hand, the lower likelihood of AI outbreaks in Africa being reported in online media, as observed in our analysis, is likely driven by a combination of factors. In some African countries, the capacity to detect, respond to, and communicate about outbreaks may be more limited, which could delay reporting and result in less online media coverage [5,33,45–48]. Furthermore, online media may prioritise specific health crises that resonate more with the local audience. In the African region, where there are often multiple ongoing health challenges (e.g., cholera, viral haemorrhagic fevers), animal outbreaks may receive less attention compared to more immediate public health threats [49,50]. Thirdly, media in some African countries may face resource constraints and infrastructure, limiting the coverage of outbreaks despite their significance [33]. These regions may also have less robust media ecosystems (including traditional and digital platforms) prioritising such events [51,52].

## Event-based surveillance from multiple sources adapted to local context

Our model results are derived from online media news of AI and ASF outbreaks, collected via PADI-web, which primarily uses the Google News aggregator as its data source [10–12]. Google News is widely recognised as one of the most prominent online news aggregators, offering extensive global coverage [53]. While Google News provide global reach, it may not fully capture news at a more local level. The inclusion of regionally focused platforms (e.g., Baidu News in China, Yandex News in Russia) may help capture outbreak-related news

of relevance to the local situation and may offer more comprehensive coverage in certain regions [54,55].

Furthermore, expert-curated EBS systems like ProMED-mail, while valuable, also exhibit regional biases, with a significant focus on countries like the United States, the United Kingdom, and China, which may receive disproportionate coverage compared to lower-income regions [56,57]. Similarly, HealthMap, like PADI-web, also relies on Google News data, along with sources such as ProMED-mail and WHO posts [58]. In addition, the Epidemic Intelligence from Open Sources (EIOS) system, a collaboration between multiple public health organisations, integrates data from several EBS systems, including HealthMap, and performs report deduplication before uploading to the platform.

Future research should aim to integrate and compare more diverse and regionally relevant data sources, including local media and community reporting, alongside global digital systems to enhance the overall effectiveness and inclusivity of EBS, particularly for lower- and middle-income countries and regions [44,59,60].

Furthermore, the news reports used in this study were collected based on English-language Google News search queries, which introduces a potential bias. Specifically, we only included news articles published in English, regardless of whether English is an official language in the country where the outbreak occurred. While this approach limited our ability to capture non-English media coverage of the AI and ASF outbreaks in 2018/19, it is important to note that PADI-web's multilingual search functionality was introduced after our analysis began. PADI-web has since broadened its language coverage, starting with seven languages in 2020 and expanding to sixteen by 2021 [12]. However, given the global dominance of English in digital media, this bias may still reflect broader trends in news coverage, with more than 25% of websites in the European Union and over 50% globally available in English [61].

Despite the limitation regarding the language search queries in PADI-web, the dataset for AI outbreaks included only 25.6% (51/199) of news reports from countries where English is an official language and even fewer (1.4%, 4/277) for ASF outbreaks. Therefore, we retained the variable indicating whether a country has English as an official language, using it as a proxy for "visibility" and international news coverage rather than considering it an artefact. In this sense, we found that only AI outbreaks in English-speaking countries were more likely to be covered by online news media, whereas this association was not observed for ASF. The observed result for AI can be partly explained by the countries with outbreaks most prominently covered in our dataset during the 2018/19 season. For example, AI outbreaks in India, the USA, and Malaysia—countries where English is either an official language or widely spoken—received substantial media attention [38,61], influencing the results observed in this analysis. Future studies should compare how outbreak reporting varies across regions and languages [48], allowing for a more comprehensive understanding of the factors influencing news coverage in the EBS space.

### Bipartite network modelling

While ERGMs have been widely used for monopartite networks in health and disease research [21,62], their use on bipartite networks is relatively recent. To fully understand the structure and dynamics of bipartite networks, we aim to explore further other methods, such as stochastic block models (SBMs, [63]), which offer a probabilistic framework for uncovering latent community structures. Additionally, matrix factorisation techniques [64] and spectral clustering [65] provide alternative approaches for analysing bipartite networks, offering insights into hidden patterns and relationships and opening new avenues for understanding how information dissemination operates in health-related contexts.

## Conclusion

This study explored the factors influencing the communication of AI and ASF outbreaks across online media sources monitored by PADI-web between 2018 and 2019. While based on a single EBS tool and two disease models, our findings show that digital outbreak communication varies between diseases, driven by the disease emergence in specific countries or regions and the involvement of specialised sources, such as national authorities, international organisations, and press agencies. These results highlight the importance of considering both the type and specialisation of sources for EI. Our analysis also highlights how the region and the language in the country of outbreaks may influence online media coverage. Future research should include a broader range of data sources, region-specific media, and multilingual news coverage to improve the inclusivity and comprehensiveness of EBS, especially in low- and middle-income countries. EI practitioners should adopt flexible, context-specific strategies for EBS tailored to the unique circumstances of each country or region. Such approaches will ensure a more accurate and thorough understanding of the disease outbreak situation, supporting better-informed and timely risk assessment and decision-making.

## Supporting information

**S1 File. Bipartite network of sources communicating on Avian Influenza and African swine fever between outbreaks 2018 and 2019 extracted with the PADI-web tool.**
**(PDF)**

**S2 File. Degree distribution for the node sources and node outbreaks of the bipartite networks for Avian influenza and African swine fever.**
**(PDF)**

**S3 File. Goodness-of-fit analysis for the Exponential Random Graph Models for Avian influenza and African swine fever.**
**(PDF)**

## Author contributions

**Conceptualization:** Pachka Hammami, Asma Mesdour, Andrea Apolloni, Elena Arsevska.

**Data curation:** Solene Rodde, Sarah Valentin, Bahdja Boudoua.

**Formal analysis:** Solene Rodde, Paulo Pimenta.

**Funding acquisition:** Elena Arsevska.

**Methodology:** Pachka Hammami, Asma Mesdour, Andrea Apolloni, Elena Arsevska.

**Resources:** Paolo Tizzani, Lina Awada, Carlene Trevennec.

**Supervision:** Pachka Hammami, Asma Mesdour, Andrea Apolloni, Elena Arsevska.

**Visualization:** Paulo Pimenta.

**Writing – original draft:** Solene Rodde, Pachka Hammami, Andrea Apolloni, Elena Arsevska.

**Writing – review & editing:** Solene Rodde, Pachka Hammami, Asma Mesdour, Sarah Valentin, Bahdja Boudoua, Paolo Tizzani, Lina Awada, Carlene Trevennec, Paulo Pimenta, Andrea Apolloni, Elena Arsevska.

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
