## [Decision Letter · Decision Letter 0]

14 Oct 2024

PONE-D-24-25259Modelling the drivers of outbreak communication from event-based surveillance for improved epidemic intelligencePLOS ONE

Dear Dr. Arsevska,

Thank you for submitting your manuscript to PLOS ONE. After careful consideration, we feel that it has merit but does not fully meet PLOS ONE’s publication criteria as it currently stands. Therefore, we invite you to submit a revised version of the manuscript that addresses the points raised during the review process.

We look forward to receiving your revised manuscript.

Kind regards,

Fernanda C. Dórea

Academic Editor

PLOS ONE

Journal Requirements:

2. Thank you for stating the following financial disclosure:H2020 MOOD project Grant agreement ID: 874850: https://cordis.europa.eu/project/id/874850

Additional Editor Comments:

Dear authors,

our sincere apologies for the delay. I took a decision to seek 3 reviewers to make sure that we were giving you a fair and balanced review. The reviewers raised important concerns related to ensuring that the utility of the tool is accurately represented, and two of the two reviewers recommended rigorous consideration (review or at least more explanations) of the statistical methods used. I ask you to please consider their comments thoughtfully and look forward to a revised version of your manuscript.

Reviewers' comments:

Reviewer's Responses to Questions

**Comments to the Author**

1. Is the manuscript technically sound, and do the data support the conclusions?

Reviewer #1: Yes

Reviewer #2: No

Reviewer #3: Yes

2. Has the statistical analysis been performed appropriately and rigorously? 

Reviewer #1: I Don't Know

Reviewer #2: No

Reviewer #3: Yes

3. Have the authors made all data underlying the findings in their manuscript fully available?

Reviewer #1: Yes

Reviewer #2: No

Reviewer #3: Yes

4. Is the manuscript presented in an intelligible fashion and written in standard English?

Reviewer #1: Yes

Reviewer #2: Yes

Reviewer #3: Yes

5. Review Comments to the Author

Reviewer #1: Overall, this manuscript is well-written and the findings highlighted will be useful to those responsible for conducting event-based surveillance using digital platforms and practitioners of epidemic intelligence.

It would be helpful if the authors could further explain the rationale for selecting news report dates restricted from July 1 2018 to June 30 2019. Were there concerns about different dynamics during the COVID-19 pandemic in 2020 and later? Were cases of AI and ASF particularly elevated at this time? Given that these data are roughly 5 years old, it would be helpful to understand why more recent data were not incorporated, as the online new environment is quite dynamic.

Given that findings that WOAH is one of the primary sources for AI and ASF, the authors might consider commenting on the implications of health authorities being the primary source vs. news agencies (e.g. what is the relative value of EBS systems picking up these reports if health authorities are already aware of the event?).

Reviewer #2: Thank you so much for submitting this work for review. There is a growing trend in the use of media aggregators to assist in media scanning efforts to inform EI and EWAR. It is great to see other tools being represented in the literature that may not be as well known as EIOS and ProMed.

With regard to this analysis, I have concerns about the subtle reference that internet-based media scanning represents a timely method of event-based surveillance to inform EWAR efforts. Media scanning is one of several other EBS modalities (media, hotline, community, and facility) that can be implemented at a country level to inform EWAR. Community is often considered the gold standard for timeliness as it picks up events even before they reach the media. Within countries, internet-based media scanning may also not be the most efficient method for tracking signals and events as much of the local media, especially social media is not currently included as a source by many of the popular aggregators. This limitation of these media aggregators can skew the ability to understand the true situation within a country or region for those not familiar with local context.

Given this stated limitation of the tool that you used, I worry that the outcomes presented in this manuscript do not truly reflect the factors/drivers of outbreak communication on a global scale and in fact give an over-representation of systems in European and English-speaking countries. It would be helpful to potentially compare this tool to other aggregators and other methods of conducting EBS to fully understand the drivers of outbreak communication globally - OR maybe a better approach would be to limit the analysis to CIRAD's global internet-based media scanning efforts and speak to how representative this is for your work and where the current gaps exist in the method's currently being used that need to be improved to gain a better global picture.

Additionally the methods you listed for collecting and verifying reports needs additional attention which may have unintentionally skewed your results as well.

I have listed more detailed comments in the attached pdf for your consideration.

Reviewer #3: Thank you for the well written article on the topic of EI. The article addresses important aspects of EI and a better understanding of EI sources can have a benefitial impact on the entire process and provides potentially helpful insights for EI practitioners. The article provides a clear description of methods and results and makes no exaggerated claims.

Some of the limitations are already discussed in the paper itself, such as the inclusion of english-only search queries and the missing of the time-aspect in the analysis. Given that, the covariate for english language should either be discarded in the analysis or intepredted differently in the results. Including the time-aspect in the analysis may provide further interesting insights.

While the usage of a bipartite network may be statistically correct, it is not clear why that is the most appropriate statistical method for the given dataset. Most of the descriptive statistics could have been achieved in a simpler way and also the computation of ORs would be possible through simple regression methods. If there are reasons for the necessity of networks, it should be made more clear.

The data and scripts is made available in a very good way and format. However, you here provide the constructed network, while more detailed information of the news articles or maybe even the articles itself are not available. This maybe benefitial for other researchers to follow up on the research.

6. PLOS authors have the option to publish the peer review history of their article (what does this mean?). If published, this will include your full peer review and any attached files.

Reviewer #1: No

Reviewer #2: No

Reviewer #3: No

---

## [Author Response · Author response to Decision Letter 1]

7 Jan 2025

Dear Editors and reviewers,

Please find attached our revised manuscript and response to your comments and review.

With best regards,

Elena Arsevska on behalf of all authors

---

## [Editor Report · Decision Letter 1]

26 Feb 2025

PONE-D-24-25259R1Modelling the drivers of outbreak communication in online media news for improved event-based surveillancePLOS ONE

Dear Dr. Arsevska,

Thank you for submitting your manuscript to PLOS ONE. After careful consideration, we feel that it has merit but does not fully meet PLOS ONE’s publication criteria as it currently stands. Therefore, we invite you to submit a revised version of the manuscript that addresses the points raised during the review process.

We look forward to receiving your revised manuscript.

Kind regards,

Fernanda C. Dórea

Academic Editor

PLOS ONE

Journal Requirements:

**Additional Editor Comments:**

Dear authors, thank you for your revised submission.

Overall, I agree with Reviewer 3 that the article provides a clear description of methods and results and makes no exaggerated claims. However, I find the discussion and conclusions still very weak. Reviewer 3 specifically asked "what is the relative value of EBS systems picking up these reports if health authorities are already aware of the event?", and I don't feel that this question was addressed in the paper or sufficiently answered. In addition, there was a question about whether the dataset was not analysed before and after the official WOAH notification, and the authors responded to this in the response letter stating that the dataset was too small to do this. But this is not reflected in the discussion, and the readers of the paper are still left wondering why this was not done. In addition to explaining why this could not be done, you should discuss what are you really able to conclude given this shortcoming, considering the question above regarding the value of EBS if the main sources are official ones. These two issues are linked.

A very small part of the discussion is focused on discussing the meaning of the results, and I don't feel it leaves readers with a clear message on the main insights learned, and how they help better implement or better use the results of EBS.

I understand your explanations for why language was not removed form the model, but the discussion seems to try to make sense of language biases as if they were a meaningful result, rather than explicitly pointing it out as an artifact of the methods. You state for example "Our results also showed that language and geography may be related". This is not a result brough to light by this work, this is just a fact. How this fact impacts the validity and should be taken into account when interpreting your results is the real information needed in the discussion.

The conclusion paragraph, in particular, gives no specific insight on what is the take home message of this paper, and what novel knowledge it provided. It states "we provide first insight that can guide practitioners when monitoring online media for EI purposes". What are those main insights and HOW are they able to guide practicioners? This point in particular - how do the insights of this paper help practicioners of EBS - is in my opinion never really made clear.

---

## [Decision Letter · Decision Letter 2]

23 Jun 2025

Modelling the drivers of outbreak communication in online media news for improved event-based surveillance

PONE-D-24-25259R2

Dear Dr. Arsevska,

We’re pleased to inform you that your manuscript has been judged scientifically suitable for publication and will be formally accepted for publication once it meets all outstanding technical requirements.

Kind regards,

Fernanda C. Dórea

Academic Editor

PLOS ONE

Additional Editor Comments (optional):

Reviewers' comments:

Reviewer's Responses to Questions

**Comments to the Author**

1. If the authors have adequately addressed your comments raised in a previous round of review and you feel that this manuscript is now acceptable for publication, you may indicate that here to bypass the “Comments to the Author” section, enter your conflict of interest statement in the “Confidential to Editor” section, and submit your "Accept" recommendation.

Reviewer #4: All comments have been addressed

2. Is the manuscript technically sound, and do the data support the conclusions?

Reviewer #4: Yes

3. Has the statistical analysis been performed appropriately and rigorously? 

Reviewer #4: Yes

4. Have the authors made all data underlying the findings in their manuscript fully available?

Reviewer #4: Yes

5. Is the manuscript presented in an intelligible fashion and written in standard English?

Reviewer #4: Yes

6. Review Comments to the Author

Reviewer #4: The authors have addressed the previous comments adequately and done an excellent job analyzing the data and presenting the results.

7. PLOS authors have the option to publish the peer review history of their article (what does this mean?). If published, this will include your full peer review and any attached files.

Reviewer #4: **Yes: **Thomas W Valente

---

## [Editor Report · Acceptance letter]

PONE-D-24-25259R2

PLOS ONE

Dear Dr. Arsevska,

I'm pleased to inform you that your manuscript has been deemed suitable for publication in PLOS ONE. Congratulations! Your manuscript is now being handed over to our production team.

Kind regards,

on behalf of

Dr. Fernanda C. Dórea

Academic Editor

PLOS ONE